# The 30-Day Economic Burden of Newly Diagnosed Complicated Urinary Tract Infections in Medicare Fee-for-Service Patients Who Resided in the Community

**DOI:** 10.3390/antibiotics11050578

**Published:** 2022-04-26

**Authors:** Thomas P. Lodise, Michael Nowak, Mauricio Rodriguez

**Affiliations:** 1Albany College of Pharmacy and Health Sciences, Albany, NY 12208, USA; 2Spero Therapeutics, Inc., Cambridge, MA 02139, USA; mnowak@sperotherapeutics.com (M.N.); mrodriguez@sperotherapeutics.com (M.R.)

**Keywords:** complicated urinary tract infections, outcomes, costs, burden of illness, Medicare

## Abstract

Introduction: Scant data are available on the 30-day financial burden associated with incident complicated urinary tract infections (cUTIs) in a cohort of predominately elderly patients. This study sought to examine total and cUTI-related 30-day Medicare spending (MS), a proxy for healthcare costs, among Medicare fee-for-service (FFS) beneficiaries who resided in the community with newly diagnosed cUTIs. Methods: A retrospective multicenter cohort study of adult beneficiaries in the Medicare FFS database with a cUTI between 2017 and 2018 was performed. Patients were included if they were enrolled in Medicare FFS and Medicare Part D from 2016 to 2019, had a cUTI first diagnosis in 2017–2018, no evidence of any UTI diagnoses in 2016, and residence in the community between 2016 and 2018. Results: During the study period, 723,324 cases occurred in Medicare beneficiaries who met the study criteria. Overall and cUTI-related 30-day MS were $7.6 and $4.5 billion, respectively. The average overall and cUTI-related 30-day MS per beneficiary were $10,527 and $6181, respectively. The major driver of cUTI-related 30-day MS was acute care hospitalizations ($3.2 billion) and the average overall and cUTI-related 30-day MS per hospitalizations were $16,431 and $15,438, respectively. Conclusion: Overall 30-day MS for Medicare FSS patients who resided in the community with incident cUTIs was substantial, with cUTI-related MS accounting for 59%. As the major driver of cUTI-related 30-day MS was acute care hospitalizations, healthcare systems should develop well-defined criteria for hospital admissions that aim to avert hospitalizations in clinically stable patients and expedite the transition of patients to the outpatient setting to complete their care.

## 1. Introduction

Complicated urinary tract infections (cUTI) are among the most frequent bacterial infections in the community and were the 14th-ranked principal diagnosis for hospital admissions in the 2018 Healthcare Cost and Utilization Project [1,2,3]. Recently, a U.S. national database study indicated that there are over 2.8 million cases of cUTI per year, resulting in annual 30-day total costs of more than $6 billion [4]. Complicated urinary tract infections are also commonplace in elderly patients [5,6,7], with a spectrum of disease severity ranging from a mild illness with limited or no systemic symptoms to severe sepsis [8,9,10,11]. Despite the frequency of cUTIs in elderly patients, most U.S. burden of illness studies have focused on younger patient cohorts [4,12] and scant data are available on the financial burden associated with incident cUTIs episodes in a cohort of predominately elderly patients. Given this gap in the literature, this study sought to examine total and cUTI-related 30-day Medicare spending (MS), a proxy for healthcare costs, among Medicare beneficiaries who resided in the community (i.e., non-long-term care facility) with newly diagnosed cUTIs. As part of the study, we were most interested in defining the number of incident cUTI cases, 30-day cUTI-related costs, and drivers of cUTI-related costs. Given that Medicare represents approximately 15% of total federal spending in the US [13], we believed a comprehensive understanding of the overall and drivers of 30-day cUTI MS would provide insights into the treatment approaches required to potentially reduce the financial burden associated with cUTIs in Medicare beneficiaries.

## 2. Methods

A retrospective multicenter cohort study of adult beneficiaries in the Medicare fee-for-service (FFS) database with a cUTI between 2017 and 2018 was performed. Patients were included in the study if they: (1) were enrolled in Medicare FFS and Medicare Part D (Medicare pharmacy benefits) from 2016 to 2019, (2) were not enrolled in Medicare Advantage, (3) had a cUTI first diagnosis (Appendix A) [4,14] in 2017–2018, (4) had no evidence of any UTI diagnoses in 2016, and (4) had no residence in a long-term care facility in 2016–2018. For patients with ≥1 cUTI between 2017 and 2018, the first cUTI was examined.

Baseline demographics and covariates included age, sex, race, dual eligibility status (enrolled in Medicare and Medicaid), and low-income subsidy (LIS) status. Charlson Comorbidity Index (CCI) score [15] and other diagnoses in the 30-day post-cUTI diagnosis period were recorded [15]. Overall and cUTI-related healthcare resource utilization (HRU), MS, and Medicare Spending per beneficiary (MSPB) were also collected in the 30-day post-cUTI index date period for the entire study population and among beneficiaries with an encounter in the following service categories: acute care inpatient facilities, non-acute/long-term care inpatient facilities, physician offices, and outpatient. Acute care inpatient facilities included acute care hospitals, critical access hospitals, and other inpatient facilities. Non-acute/long-term care inpatient facilities included skilled nursing facilities (SNF), inpatient rehabilitation facilities (IRF), long-term acute care hospitals (LTACH), and inpatient psychiatric facilities. Physician classifications included primary care, physician specialist, and carrier non-physicians. Outpatient included hospital outpatient, home health, hospices, renal dialysis, outpatient nursing homes, outpatient pharmacy (Medicare Part D), and other outpatient. To quantify the 30-day economic burden of newly diagnosed cUTIs in Medicare beneficiaries who resided in the community, the following overall and cUTI-related outcomes were reported: (1) MS (total sum of MS in the 30-day follow-up period across all beneficiaries included in the study), (2) MS in each service category (total sum of MS in each service category in the 30-day follow-up period across all beneficiaries included in the study), (3) average MSPB (total sum of MS divided by the number of beneficiaries included in the study), (4) average number of encounters per beneficiary (total number of encounters in the study population divided by number of beneficiaries included in the study) (5), average number of encounters per beneficiary in each service category (total number of encounters in the study population in each service category divided by the number of beneficiaries included in the study), and (6) average MS per encounter in each service category (total sum of MS in the study population in each service category divided by the number of total encounters in the service category). Medicare spending was considered related to the cUTI if the claim(s) included any diagnosis (i.e., primary or secondary) of a UTI (Appendix A) [4]. Data aggregation, analysis, and visualization were performed using Tableau 2021.4 and Microsoft Excel.

## 3. Results

Among the beneficiaries in Medicare FFS only and Medicare Part D between 2016 and 2019 with no evidence of any UTI diagnoses in 2016, 2,330,123 had a newly diagnosed UTI in 2017–2018. Of the 2,330,123 beneficiaries with a UTI, 772,896 had a cUTI and 723,324 (93.5%) occurred in beneficiaries who resided in the community (i.e., non-long-term care facility). Among the beneficiaries in the final study population, most were male (61%), and age distribution was as follows: 13% were <65 years, 29% were 65–74 years, 35% were 75–84 years, and 23% were ≥85 years. Seventeen percent of beneficiaries in the study population had partial or full dual Medicare and Medicaid coverage and 19% had a partial or full LIS status (Table 1). The average CCI score was 2.2 in the 30-day post-cUTI index follow-up period. Diagnosed conditions identified in the 30-day post-cUTI index follow-up period are listed in Appendix A.

Overall and cUTI-related 30-day MS were $7.6 and $4.5 billion, respectively (Figure 1). The average overall and cUTI-related 30-day MSPB were $10,527 and $6181, respectively. Acute care hospitalizations represented the highest proportion of overall and cUTI-related MS and MSPB. Non-acute/long-term care facilities were the second highest, followed by physician, and outpatient services. On average, beneficiaries had 4.4 and 2.2 overall and cUTI-related service encounters, respectively. The mean number of encounters per beneficiary in each service category and average MS per encounter in each service category are displayed in Figure 2. The mean number of encounters per beneficiary for cUTI-related physician specialists and primary care visits were 0.8 and 0.3, respectively. Receipt of care in acute care and outpatient hospitals were also commonplace in the 30-day follow-up period (mean number of cUTI-related encounters per beneficiary was 0.3 for both service categories). The mean number of cUTI-related encounters per beneficiary was ≤0.05 for all other service categories. Average MS for cUTI-related encounters for physician specialist, primary care, and outpatient hospitals were $269, $310, and $869, respectively. Among beneficiaries with acute care hospitalizations, the average overall and cUTI-related 30-day MS per encounter were $16,431 and $15,438, respectively.

## 4. Discussion

Thirty-day MS for beneficiaries who resided in the community (i.e., non-long-term care facility) with incident cUTIs between 2017 and 2018 was substantial (overall: $7.6 billion), with cUTI-related MS accounting for 59% (cUTI-related: $4.5 billion) of 30-day MS. On average, 30-day cUTI-related MSPB was $6181. The major driver of 30-day cUTI-related MS was acute care hospitalizations, which totaled at $3.2 billion. Approximately 30% of beneficiaries in the study cohort had an acute care hospitalization and the average cost of a cUTI-related hospitalization was $15,438. Although the proportion of beneficiaries with encounters for non-acute long-term care facilities in the 30-day follow-up period was low, it was the second major contributor to 30-day cUTI-related MS, followed by MS on physician visits and other outpatient medical services. Medicare Part D spending (cUTI-related: 7.6 million) was only a small component of 30-day cUTI-related MS, reflecting the low costs of the generic antibiotics used to treat cUTIs patients.

To put these 30-day cUTI-related MS statistics in proper perspective, the annual Medicare FFS payments for Parts A and B benefits were $403 billion, and $95 billion for Part D, in 2018 [13]. Annual average MS per enrollee in Medicare Parts A and/or B was $10,229 in 2018 [16]. Thus, the cUTI-related MS in the 30-day post-cUTI follow-up period alone consumed ~0.5% of the annual Medicare budget FFS for Parts A and B, but was only a small fraction of Part D spending. More importantly, 30-day cUTI-related MS for each beneficiary who experienced a cUTI was considerable when compared with the total average amount spent on each beneficiary per year. Given the aging of the population and growth in Medicare enrollment, the future number of cUTIs and total cUTI-related MS is expected to be even greater. These findings highlight the critical need for educating patients on ways to mitigate the occurrence of cUTIs (i.e., maintain adequate hydration, proper hygiene, urinate as soon as the need arises, etc.). They also indicate that clinicians need to proactively implement cUTI prevention measures [17,18], especially among patients with indwelling urinary catheters, and create individualized patient care plans [19] to reduce the clinical and economic sequelae associated with cUTIs. Since 70% of cUTI-related 30-day MS was for acute care hospitalizations, healthcare systems also need to develop institutional site-of-care clinical pathways that aim to expedite the transition of patients to the outpatient setting once they are stabilized in the emergency department, observation unit, or hospital [20,21,22,23,24]. Given the vulnerability of many elderly patients and observed diagnosed conditions identified in the 30-day post-cUTI index follow-up period (Appendix A), it is likely that most elderly patients will require initial care in the hospital. However, data indicates that admission patterns were highly variable among cUTI patients and many hospitalized cUTI patients ≥65 years had low disease acuity and were potential candidates for outpatient care or early hospital discharge [4,12,25]. For instance, if hospitals were able to avert acute care hospitalizations by 5–10% among elderly patients, it would reduce 30-day cUTI MS by upwards of $200–400 million.

Outpatient treatment options for appropriate elderly cUTI patients that may potentially lessen the financial burden of cUTI-related hospitalization on the Medicare program include oral antibiotics and outpatient parenteral therapy (OPAT). While it is always preferred to treat with oral antibiotics when possible, their use is somewhat limited for many elderly cUTI patients due to the high resistance rates to first-line oral cUTI agents among common uropathogens [26,27,28]. An alternative to oral antibiotics is OPAT, which can be administered at home, in a physician’s office or infusion suite, or in a non-acute long-term care facility. There are several considerations with the use of OPAT across these settings. Medicare currently provides limited coverage of home infusions, limiting their use for many elderly cUTI patients [29]. It is challenging for many elderly patients, especially those homebound, to travel daily to physicians’ offices/infusion centers for receipt of OPAT. Furthermore, data indicate that elderly patients are at an increased risk for OPAT-related adverse events and subsequent hospital admissions/readmissions [30,31,32,33,34,35,36,37,38]. Cost of delivering care in non-acute long-term care facilities is comparable to those associated with acute care hospitalizations [39]. Although their use was limited in the 30-day post-cUTI follow-up period, the average MS for a SNF, IRF, and LTACH encounter were $8859, $23,401, and $41,127, respectively. Fortunately, there are several pharmacokinetic and pharmacodynamic dose-optimized oral antibiotics in development with reliable activity against highly resistant common uropathogens [40] and these have the potential to lessen the financial burden of cUTIs on the Medicare program. Like all new treatments, the ability of these new agents to improve the efficiency of healthcare delivery for elderly cUTI patients will need to be demonstrated before they can be adopted in clinical practice.

Several things should be noted when interpreting study findings. This study was subject to the limits inherent in all administrative claims database analyses. Diagnoses of cUTIs were based on diagnostic and procedure codes, but many of the claims used have been shown to have high positive predictive values [41,42]. The study population included all Medicare FFS beneficiaries with part D coverage, including those ≤65 years who qualified for Medicare due to a permanent disability. We included Medicare beneficiaries <65 years because they represented 12.9% of the study population and their rates of chronic conditions, functional limitations, and cognitive impairments are consistent with Medicare beneficiaries ≥65 years, which minimizes the potential for any downward bias on observed results due to younger age [43]. Due to the nature of the study, we could not determine if beneficiaries presented to the hospital with a cUTI or developed their cUTI during a hospitalization. However, it is likely that a fair proportion of patients presented to the hospital with a cUTI given that this report was limited to beneficiaries who resided in the community. Lastly, MS may have been a conservative estimate of the true costs as data shows that the amount reimbursed by Medicare is often less than the accrued healthcare costs [44]. Future studies are needed to determine the actual cost of cUTI-related hospitalizations. Since cUTIs have different pathologies, future studies should analyze costs across the different cUTI diagnosis-related groups (DRGs).

In summary, 30-day MS for beneficiaries who resided in the community (i.e., non-long-term care facility) with incident cUTIs was substantial, with cUTI-related MS accounting for 56% of the total. While a variety of service categories contributed to 30-day cUTI-related MS, the major driver was acute care hospitalizations. Healthcare systems should develop well-defined criteria for hospital admissions that aim to expedite the transition of stable patients to the outpatient setting to complete their care. Although acute care hospitalization will still be required for most elderly cUTI patients, even modest reductions in hospitalization rates will have a major impact on cUTI-related MS. The findings also highlight the need for additional treatment options that maximize the efficiency of healthcare delivery for beneficiaries who can be safely and effectively managed in the outpatient setting.

## Figures and Tables

**Figure 1 antibiotics-11-00578-f001:**
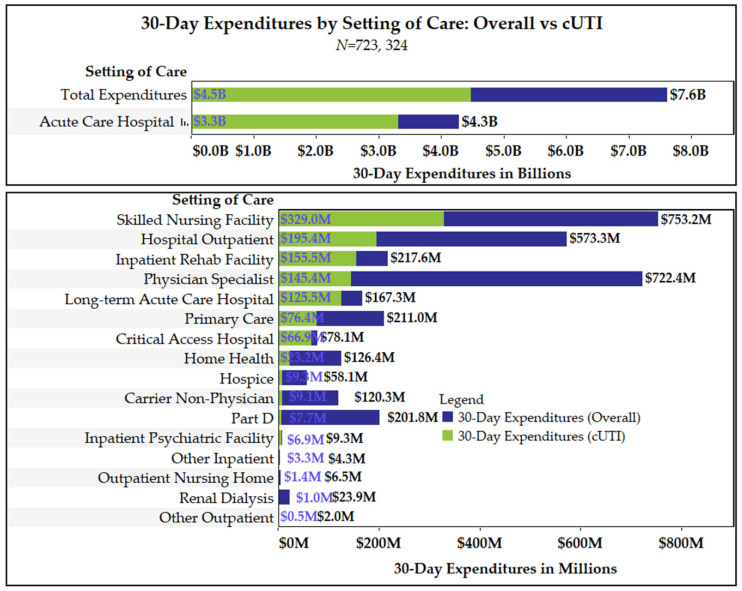
Overall and cUTI-related 30-day Medicare spending by setting of care.

**Figure 2 antibiotics-11-00578-f002:**
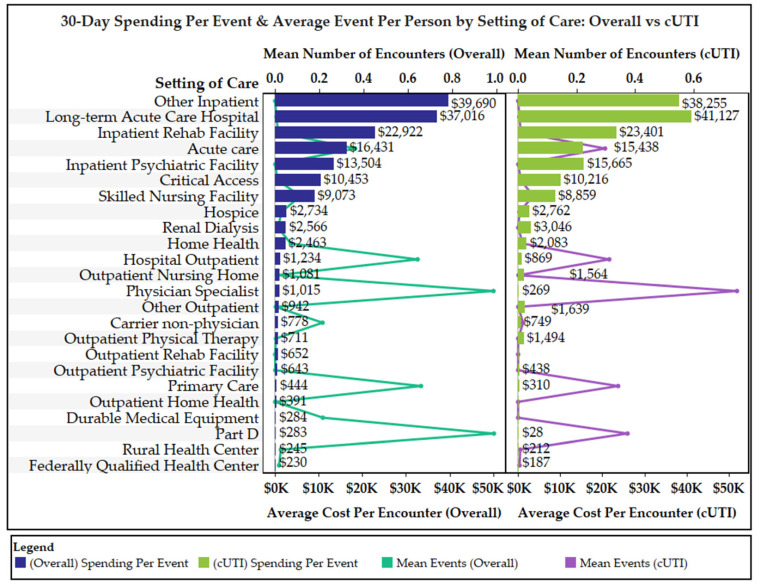
Mean number of overall and cUTI-related 30-day encounters per beneficiary in each service category and average 30-day Medicare spending per encounter in each service category.

**Table 1 antibiotics-11-00578-t001:** Baseline demographic and clinical characteristics.

Characteristics	Patients	
	(N = 723,324)	
Age Distribution	*n*	(%)
<65	93,459	(12.9%)
65–74	211,279	(29.2%)
75–84	253,524	(35.0%)
85+	165,062	(22.8%)
Sex		
Male	441,452	(61.0%)
Female	256,878	(35.5%)
Unknown	24,994	(3.5%)
Race		
Non-Hispanic White	575,316	(79.5%)
Black	49,713	(6.9%0
Hispanic	39,770	(5.5%)
Asian/Pacific Islander	17,163	(2.4%)
Other/Unknown	41,362	(5.7%)
Dual eligible status		
Full dual	95,311	(13.2%)
Partial dual	27,145	(3.8%)
Non-dual	600,868	(83.1%)
Low-income subsidy (LIS) status		
Full LIS	124,354	(17.2%)
Partial LIS	14,604	(2.0%)
No LIS	584,366	(80.8%)

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
