# Peer review of "The 30-Day Economic Burden of Newly Diagnosed Complicated Urinary Tract Infections in Medicare Fee-for-Service Patients Who Resided in the Community"

_antibiotics, 2022, doi:10.3390/antibiotics11050578_

Round 1

Reviewer 1 Report

Dear authors,

First of all, congratulations to the authors for the manuscript. The use of Medicare (MS) spending as a proxy for health care costs has been interesting to me. Regarding the manuscript, I think the development is adequate. The introduction is clear and concise. The methodology is also adequate, although it is suggested that the way of obtaining the databases used should be detailed. The results are described correctly.

In general, the manuscript is adequate. The only thing that is not clear to me is the delimitation of the analyzed sample, I do not know when it refers to the community, I do not know exactly what the group is delimited, this should be clarified. Is it a cut of all adult patients with a cUTI between 2017 and 2018? What is its geographic reach? The discussion is interesting, in addition the limitations of this type of study are detailed, therefore, as a suggestion, I would recommend including at the end, that it would be necessary to carry out more studies in greater detail to know more accurately the real cost of hospitalization. Starting from the basis that not all patients behave in the same way during hospitalization even if they have the same pathology. For this reason, for example, it would be interesting to analyze the cost through the groups related to the diagnosis (DRG).

In general, I find the approach used interesting. 

Author Response

Reviewer 1

First of all, congratulations to the authors for the manuscript. The use of Medicare (MS) spending as a proxy for health care costs has been interesting to me. Regarding the manuscript, I think the development is adequate. The introduction is clear and concise. The methodology is also adequate, although it is suggested that the way of obtaining the databases used should be detailed. The results are described correctly.

  • Thank you for the positive feedback.

In general, the manuscript is adequate. The only thing that is not clear to me is the delimitation of the analyzed sample, I do not know when it refers to the community, I do not know exactly what the group is delimited, this should be clarified. Is it a cut of all adult patients with a cUTI between 2017 and 2018?

  • Recommendation incorporated. Community indicates patients who did not reside in a long-term care facility.  We now make this clearer throughout the manuscript.  In every instance we state “community”, we now include “community (i.e., non- long-term care facility).”
  • The data reflects all patients enrolled in Medicare FFS and Medicare Part D from 2016-2019 and not enrolled in Medicare Advantage with a cUTI first diagnosis in 2017-2018. Because of confidentiality, they only provided us with the number of beneficiaries in Medicare FFS only and Medicare Part D with an UTI.

What is its geographic reach? The discussion is interesting, in addition the limitations of this type of study are detailed, therefore, as a suggestion, I would recommend including at the end, that it would be necessary to carry out more studies in greater detail to know more accurately the real cost of hospitalization. Starting from the basis that not all patients behave in the same way during hospitalization even if they have the same pathology. For this reason, for example, it would be interesting to analyze the cost through the groups related to the diagnosis (DRG).

  • The data reflects all beneficiaries in Medicare FFS only and Medicare Part D between 2016-2019 throughout the United States.
  • Recommendation incorporated. The following was added to the manuscript, “Future studies are needed to determine the actual cost of cUTI-related hospitalizations. Since cUTIs have different pathologies, future studies should analyze costs across the different cUTI diagnosis-related groups (DRGs).”

Reviewer 2 Report

I have some little recommendations from my point of view

  1. In the introduction section authors could explain the importance of the study, subject and the problem more precisely.
  2. The patient characteristics and the main variables used in the study are well presented and understandable. It is not absolutely clear how the sum $7.6 and $4.5 billion were calculated exactly. I think that illustration of categories in each setting will be more illustrative and interesting as it could make visible the burden of each subdivision. It could be added in the methodology section as a short explanation also.
  3. It would be interesting if the authors discussed similar results presented in other studies considering the cUTI-related resource utilization in other countries.
  4. The other aspect that could be commented on is the value of treatment, especially the medicines used (generics or branded). What is their impact on costs as a part of overall costs?
  5. I can't find in the text where the supplemental table 1 is mentioned? I would like to note that this table is not understandable. 

Author Response

Reviewer 2.

In general, I find the approach used interesting.

  • Thank you for the positive feedback.

I have some little recommendations from my point of view

In the introduction section authors could explain the importance of the study, subject and the problem more precisely.

  • Recommendation incorporated. The following was added to the Introduction, “As part of the study, we were most interested in defining the number of incident cUTI cases, 30-day cUTI-related costs, and drivers of cUTI-related costs.  Given that Medicare represents approximately 15% of total federal spending in the US [13], we believed a comprehensive understanding of the overall and drivers of 30-day cUTIs spending would provide insights on the treatment approaches required to potentially reduce the financial burden associated with cUTIs in Medicare beneficiaries.”

The patient characteristics and the main variables used in the study are well presented and understandable. It is not absolutely clear how the sum $7.6 and $4.5 billion were calculated exactly. I think that illustration of categories in each setting will be more illustrative and interesting as it could make visible the burden of each subdivision. It could be added in the methodology section as a short explanation also.

  • Recommendation incorporated. The outcomes section now reads, “To quantify the 30-day economic burden of newly diagnosed cUTIs in Medicare beneficiaries who resided in the community, the following overall and cUTI-related outcomes were reported: (1) MS (total sum of MS in the 30-day follow-up period across all beneficiaries included in the study), (2) MS in each service category (total sum of MS in each service category the 30-day follow-up period across all beneficiaries included in the study), (3) average MSPB (total sum of MS divided by number of beneficiaries included in the study), (4) average number of encounters per beneficiary (total number of encounters in the study population divided by number of beneficiaries included in the study) (5), average number of encounters per beneficiary in each service category (total number of encounters in the study population in each service category divided by number of beneficiaries included in the study), and (6) average MS per encounter in each service category (total sum of MS in the study population in each service category divided by the number of total encounters in the service category).  Medicare spending was considered related to the cUTI if the claim(s) included any diagnosis (i.e., primary or secondary) of a UTI (Supplemental Table 1) [4].”
  • We also revised figures 1 and 2 based on your query.

It would be interesting if the authors discussed similar results presented in other studies considering the cUTI-related resource utilization in other countries.

  • We agree with the reviewer that it would be interesting to compare results to other similar studies. Unfortunately, we performed a literature search and could not finding a comparable study conducted in a different country.

The other aspect that could be commented on is the value of treatment, especially the medicines used (generics or branded). What is their impact on costs as a part of overall costs?

  • We only had access to total amount spend on Medicare Part D (prescription benefits). We include the following in the discussion, “Medicare Part D spending (cUTI-related: 7.6 million USD) was only a small component of 30-day cUTI-related MS, reflecting the low costs of the generic antibiotics used to treatment cUTIs patients.”
  • We now make it clearer in the methods that Medicare Part D is the pharmacy benefit part of Medicare.       

I can't find in the text where the supplemental table 1 is mentioned? I would like to note that this table is not understandable.

  • Supplemental table 1 is now more clearly defined in the methods. it is the 

    Reviewer 2.

    In general, I find the approach used interesting.

    • Thank you for the positive feedback.

    I have some little recommendations from my point of view

    In the introduction section authors could explain the importance of the study, subject and the problem more precisely.

    • Recommendation incorporated. The following was added to the Introduction, “As part of the study, we were most interested in defining the number of incident cUTI cases, 30-day cUTI-related costs, and drivers of cUTI-related costs.  Given that Medicare represents approximately 15% of total federal spending in the US [13], we believed a comprehensive understanding of the overall and drivers of 30-day cUTIs spending would provide insights on the treatment approaches required to potentially reduce the financial burden associated with cUTIs in Medicare beneficiaries.”

    The patient characteristics and the main variables used in the study are well presented and understandable. It is not absolutely clear how the sum $7.6 and $4.5 billion were calculated exactly. I think that illustration of categories in each setting will be more illustrative and interesting as it could make visible the burden of each subdivision. It could be added in the methodology section as a short explanation also.

    • Recommendation incorporated. The outcomes section now reads, “To quantify the 30-day economic burden of newly diagnosed cUTIs in Medicare beneficiaries who resided in the community, the following overall and cUTI-related outcomes were reported: (1) MS (total sum of MS in the 30-day follow-up period across all beneficiaries included in the study), (2) MS in each service category (total sum of MS in each service category the 30-day follow-up period across all beneficiaries included in the study), (3) average MSPB (total sum of MS divided by number of beneficiaries included in the study), (4) average number of encounters per beneficiary (total number of encounters in the study population divided by number of beneficiaries included in the study) (5), average number of encounters per beneficiary in each service category (total number of encounters in the study population in each service category divided by number of beneficiaries included in the study), and (6) average MS per encounter in each service category (total sum of MS in the study population in each service category divided by the number of total encounters in the service category).  Medicare spending was considered related to the cUTI if the claim(s) included any diagnosis (i.e., primary or secondary) of a UTI (Supplemental Table 1) [4].”
    • We also revised figures 1 and 2 based on your query.

    It would be interesting if the authors discussed similar results presented in other studies considering the cUTI-related resource utilization in other countries.

    • We agree with the reviewer that it would be interesting to compare results to other similar studies. Unfortunately, we performed a literature search and could not finding a comparable study conducted in a different country.

    The other aspect that could be commented on is the value of treatment, especially the medicines used (generics or branded). What is their impact on costs as a part of overall costs?

    • We only had access to total amount spend on Medicare Part D (prescription benefits). We include the following in the discussion, “Medicare Part D spending (cUTI-related: 7.6 million USD) was only a small component of 30-day cUTI-related MS, reflecting the low costs of the generic antibiotics used to treatment cUTIs patients.”
    • We now make it clearer in the methods that Medicare Part D is the pharmacy benefit part of Medicare.       

    I can't find in the text where the supplemental table 1 is mentioned? I would like to note that this table is not understandable.

    • Supplemental table 1 is now more clearly defined in the methods. it is the published algorithm, based on ICD-10 diagnosis codes, to define patients with cUTIs.
    • The title of supplemental table 1 now reads, “Algorithm for Identifying Medicare Beneficiaries with cUTIs Based on International Classification of Diseases, Tenth Revision (ICD-10) Diagnostic Codes and Procedure Codes and Current Procedural Terminology (CPT) Codes.”
    published algorithm, based on ICD-10 diagnosis codes, to define patients with cUTIs.
  • The title of supplemental table 1 now reads, “Algorithm for Identifying Medicare Beneficiaries with cUTIs Based on International Classification of Diseases, Tenth Revision (ICD-10) Diagnostic Codes and Procedure Codes and Current Procedural Terminology (CPT) Codes.”
